

# Environmental risks and agronomic benefits of industrial sewage sludge-derived biochar

Vladimír Fришták[1], Lucia Polt'áková[1], Gerhard Soja[2], Hana Kaňková[3], Katarína Ondreičková[4], Elena Kupcová[5] and Martin Pipíška[1]

[1] Department of Chemistry, University of Trnava, Trnava, Slovakia
[2] Institute for Chemical and Energy Engineering, Universität für Bodenkultur Wien, Vienna, Austria
[3] FunGlass, Centre for Functional and Functionalized Glass, University of Trenčín, Trenčín, Slovakia
[4] Research Institute of Plant Production, National Agricultural and Food Centre, Piešťany, Slovakia
[5] Department of Chemistry, Faculty of Natural Sciences, Matej Bel University in Banská Bystrica, Banská Bystrica, Slovakia

Corresponding author
Vladimír Fришták,
vladimir.fristak@truni.sk

## ABSTRACT

The main objective of the present work was to assess the ecotoxicological safety of the use of thermochemically treated sewage sludge from the wastewater treatment plant (WWTP) of a distillery plant as a soil additive in agricultural soils based on its physicochemical characteristics and the bioaccumulation of selected elements in the plant tissues of maize (*Zea mays*). We have carried out physicochemical characterization (pH, EC, $C_{org}$, $C_{inorg}$, CEC, N, H, ash content, PAHs) of sewage sludge feedstock (SS) and sludge-derived biochar (BC) produced by slow pyrolysis at a temperature of 400 °C. The effect of 1% (w/w) amendment of SS and BC on soil physicochemical properties (pH, EC, $C_{inorg}$), germination of ryegrass, soil rhizobacteria and microorganisms, as well as on the accumulation and translocation of selected elements in maize (*Zea mays*) was studied. The results show that pyrolysis treatment of distillery WWTP sludge at 400 °C increases pH (from 7.3 to 7.7), $C_{org}$ (from 28.86% to 36.83%), N (from 6.19% to 7.53%), ash content (from 23.59% to 50.99%) and decreases EC (from 2.35 mS/cm to 1.06 mS/cm), CEC (from 118.66 cmol/kg to 55.66 cmol/kg), H (from 6.76% to 1.98%) and Σ18 PAHs content (from 4.03 mg/kg to 3.38 mg/kg). RFA analysis of SS and BC showed that pyrolysis treatment multiplies chromium (Cr) (2.2 times), nickel (Ni) (2.96 times), lead (Pb) (2.13 times), zinc (Zn) (2.79 times), iron (Fe) (1.26 times) in the obtained BC, but based on an ecotoxicological test with earthworms *Eisenia fetida*, we conclude that pyrolysis treatment reduced the amount of available forms of heavy metals in BC compared to SS. We demonstrated by a pot experiment with a maize that a 1% addition of BC increased soil pH, decreased EC and $C_{inorg}$ and had no significant effect on heavy metal accumulation in plant tissues. According to the results of the three-level germination test, it also does not affect the germination of cress seeds (*Lepidium sativum*). There was a significant effect of 1% BC addition on soil microbial community, and we observed a decrease in total microbial biomass and an increase in fungal species variability in the soil. Based on these results, we conclude that BC represents a promising material that can serve as a soil additive and source of nutritionally important elements after optimization of the pyrolysis process.

# INTRODUCTION

Globally, increasing industrial activity and human population growth bring with them a range of negative impacts on the environment. One of these is the production of bulk sludge as a by-product of municipal and industrial water treatment in wastewater treatment plants (WWTPs). The handling, recovery and disposal of these wastes represents a challenge that must be addressed from a sustainability perspective. Conventional treatment practices for sewage sludge are incineration and landfilling respectively. Agronomical applications of sludge are very rare in terms of current EU legislation. On the other hand, sludge is a relatively valuable material, rich in an organic matter and nutritionally important elements (carbon (C), nitrogen (N), phosphorus (P), zinc, (Zn), *etc.*). Therefore, the possibility of alternative treatment is constantly being sought, while maintaining the above-mentioned benefits and eliminating the risks of untreated sludge application to the soil. These include the danger of microbial contamination and the penetration of inorganic (heavy metals) and organic contaminants (polycyclic aromatic hydrocarbons—PAHs, polychlorinated dibenzo-p-dioxins and dibenzofurans—PCDD/Fs, polychlorinated biphenyls—PCBs and microplastics) into the soil ecosystem.

Pyrolysis treatment of sludge produced by industrial and municipal WWTPs represents a relatively innovative and promising method for the efficient use of waste material to obtain a valuable product rich in organic carbon. Several scientific studies (*Frišták et al., 2022*; *Xiao et al., 2022*; *Zhou et al., 2023*; *Wan et al., 2024*) point to the effectiveness of pyrolysis treatment of sludge in terms of its hygienization (elimination of pathogenic microorganisms) and reduction of the risk of accessibility of contaminants to living organisms. According to *Chen et al. (2015)*, pyrolysis sludge treatment represents a sustainable solution for the treatment of sewage sludge from different input sources, as it solves the issues of energy recovery, nutrient recycling, immobilization of heavy metals, as well as overall environmental protection. *Gao et al. (2017)* reported in their work that sludge pyrolysis converts approximately 50% of the sludge mass into biochar, which can then be used as a solid fuel, adsorbent of heavy metals and organic pollutants, catalyst, or as a soil additive. However, as we know very well, soil applications and the overall handling of sludge and sludge-based pyrolysis products are limited by several factors (pyrolysis temperature, sludge pre-treatment, residence time, *etc.*). Pyrolysis is a complex multi-step process in which organic matter is thermally decomposed by increasing temperature (*Rumaihi et al., 2022*). Different perspectives on the phases of sludge pyrolysis suggest that these differences may be due to the diverse experimental conditions and heterogeneous properties of the clarified sludge. *Manara & Zabaniotou (2012)* reported in their work that low temperatures are inefficient for decomposition of organic matter, and higher temperatures, although providing sufficient energy for secondary pyrolysis, reduce the biochar yield from sludge. During the pyrolysis process, most of the heavy

metals are not eliminated from the sludge but transformed into a different chemical form *Praspaliauskas, Pedišius & Striugas (2018)*. Metals occurring in the form of mineral salts and hydroxides are usually transformed into oxides or sulphides with higher thermal stability. The consequence of this transformation is a decrease in the mobility of metals in the resulting sludge biochar (*Lu et al., 2016*). According to *Li et al. (2018)*, the main reasons for the immobilization of heavy metals in biochar are related to the formation of new stable crystalline forms of carbon , which absorb heavy metals through their reaction with elements in minerals, substitution of mineral elements, and occupation of vacancies in minerals. Biochar, in a figurative sense, can fix heavy metals within its structure, leading to their lower leachability and release into the environment.

Sludge-based biochar has the potential to serve as an organic fertiliser, which is an important source of essential elements and microelements for plants. Also, the material characteristics of biochar such as pH and porosity contribute to the retention of some nutritionally important elements and soil moisture. Among other things, sludges contain different concentrations and forms of heavy metals, which are multiplied in the structure of the resulting biochar during pyrolysis treatment.

Based on the above knowledge, in the present work we decided to characterize the pyrolysis product based on the sewage sludge from the industrial wastewater treatment plant of a distillery in terms of bioavailability of the selected microelements and potentially toxic elements. The aim of the work was the imaginary closure of the cycle: useful plant—harvesting—industrial processing—sludge production—sludge processing—fertilizer production for the cultivation of the useful plant.

## MATERIALS & METHODS

### Sewage sludge origin and biochar production

A sample of the distillery's WWTP sludge was obtained after centrifugation of the return sludge from the collection tank of Enviral, a.s. (Leopoldov, Slovakia), a company involved in bioethanol production. The sludge obtained was oven-dried at 50 °C to constant weight. Part of the dried sludge was thermochemically treated by a slow pyrolysis process at 400 °C in a continuous pyrolysis reactor PYREKA (Pyreg, Grevesmühlen, Germany) under strictly anoxic conditions. Slow pyrolysis was carried out at a heating rate of 30 °C/min and after reaching the maximum temperature the feed material was kept in the reactor for 20 min. The use of gaseous $N_2$ with a flow rate of 2 L/min of the reactor exchange volume ensured that a strictly anoxic atmosphere was achieved. The sewage sludge- derived biochar (BC) and dried sewage sludge (SS) were sieved to a fraction of 0.25–2 mm using standard sieves.

### Physico-chemical characterization of sewage sludge-derived biochar

The SS and BC samples were exposed to basic physicochemical characterization. All determinations were carried out in triplicates. For pH determination the KCl method was used. The BC and SS samples were mixed with 1 mol/L KCl at volume ratio 1/10 (v/v). The pH was measured using a pH/EC/TDS multimeter HI5521 (Hanna Instruments). The determination of electrical conductivity (EC) was carried out by mixing SS and BC samples with deionized water at ratio 1/10 (v/v) and measured using a pH/EC/TDS

multimeter HI5521 (Hanna Instruments). To quantify the carbonate content in the BC and SS samples, the Jankov calcimeter was used. The analysis of cation exchange capacity (CEC) was carried out using the method modified by *Frišták et al. (2013)* in three repetitions. The $Ba^{2+}$ as a loading cation and $Mg^{2+}$ as an exchanging cation were used. A modified toluene extraction procedure according to *Hilber et al. (2017)* was used to determine the total concentration of 18 extractable structures of PAHs (polycyclic aromatic hydrocarbons). The obtained extracts were concentrated using a Hei-Vap vacuum evaporator (Heidolph, Scwabach, Germany). For the quantification of 18 structures of PAHs compounds namely naphthalene, acenaphthylene, acenaphthalene, fluorene, phenanthrene, anthracene, fluoranthene, pyrene, benzo(a)anthracene, chrysene, benzo(b)fluoranthene, benzo(a)pyrene, dibenzo(a,h)anthracene, benzo(g,h,i)perylene, ideno(1,2,3)pyrene, 1-methylnaphtalene and 2-methylnaphthalene, high-performance liquid chromatography HPLC (1260 Infinitym AGILENT Technologies) was employed. Total concentrations of carbon (C), hydrogen (H) and N in BC and SS samples were quantified by CHNS-O elemental analyser (CHNS-O EA 1108; Carlo Erba Instruments, Milan, Italy). For the determination of total concentrations of arsenic (AS), cadmium (Cd), chromium (Cr), copper (Cu), nickel (Ni), lead (Pb), Zn, iron (Fe), and mercury (Hg), the method of X-ray fluorescence spectrometry (RFA) was used by Spectro Xepos-XRF meter (Spectro AnalyticL Instruments GmbH, Kleve, Germany).

## Short-term ecotoxicological tests

The short-term germination test (7 days) with cress seeds (*Lepidium sativum*) was performed according to the scheme described in our previous work (*Frišták et al., 2022*). Nine testing containers were prepared for the test, three serving as a control, three containing 1% (w/w) amendment of BC and three containing 1% (w/w) amendment of SS. Washed and chemically pure sand served as a medium. The containers were divided into three levels and the effects of contaminants (gaseous, water-soluble, and bound to BC and SS) on the germination of cress seeds were studied. At the first level, the effect of volatile compounds released from BC and SS materials on germination was studied. At the second level, the effect of gaseous contaminants, water-soluble contaminants, and direct contact of seeds with BC and SS materials was studied. At the third level, the effect of volatile compounds and water-soluble compounds from BC and SS materials on seed germination was studied. To assess the suitability and toxicological safety of the analyzed BC and SS samples as soil additives, a long-term ecotoxicological test (120 days) using selected individuals of the California earthworm (*Eisenia fetida*) obtained from a vermicomposting station with certified origin was performed. Containers with conventional commercially available garden substrate (Natura, Agro CS, Slovakia, pH 6.8, made only from natural raw materials: peat, coconut peat and clay, with a modified limestone reaction) were amended with 1% (w/w) addition of SS and 1% (w/w) addition of BC. Containers with non-amended substrate represented control. In each container 10 adult earthworm specimens were added. The soil moisture was monitored by moisture sensors EcoWitt WH51 (EcoWitt, Shenzhen, China) and adjusted between 50 and 60% during experimental period. Once a month the counting abundance of earthworms, measurement of size and application of a feeding

ration consisting of coffee grounds, carrots and leaf lettuce (3:2:1 ratio) were done. Feed rations were always applied in equal amounts to each experimental container and manually incorporated to a maximum of two cm depth of substrate. After 120 days the specimens were left emptied and executed by ethanol method. Tissues were digested by a microwave assisted digestion system (Multiwave GO PLUS; Antoon Paar, Graz, Germany) with $HNO_3$ and characterize by spectral analysis for the determination of selected heavy metals by atomic absorption spectrometry with electrothermal atomization (ET—AAS) and with flame atomization (F—AAS) using an Agilent 240FS AA spectrometer (Agilent).

## Pot experiment with *Zea mays* L.

To assess how the BC and SS samples as soil additives affect plant growth and metal accumulation in the plant, a long-term outdoor pot experiment with maize (*Zea mays* L.) was carried out. The arable soil used as a substrate in the experimental pots was collected at the site of the village of Zákamenné in North of Slovakia (49°24′44.3″N19°17′14.6″E) in April 2023 where the experiment was set up. After collection, the substrate was air-dried, homogenized, stripped of plant and animal residues and sieved to a fraction >two mm using standard sieves. Subsequently, nine containers were applied, three serving as control (K1-3), three containing substrate with 1% (w/w) addition of BC (BC1-3) and three containing substrate with 1% (w/w) addition of SS (SS1-3). Preparation of the containers consisted of creating a drainage layer consisting of mesh and gravel, weighing the substrate and 1% additions, which were thoroughly mixed into the substrate. Maize seeds were soaked in rainwater for 3 days before being placed in the containers, to identify damaged seeds and to accelerate germination. Subsequently, into each pot were placed 6 seeds, which were allowed to germinate, and after germination, just one individual plant in each pot was left. On the day of planting, the substrates were moistened with rainwater to ensure sufficient moisture for germination. The randomized experimental set-up of pots has been used. External environmental conditions such as daily temperature, humidity and volume of rainfall were monitored throughout the experiment. The experiment was finished after twenty-two weeks by harvesting the plant objects as well as taking representative soil substrate samples for subsequent chemical and microbiological analysis.

## Physico-chemical and microbiological analysis of plant and soil samples

Obtained maize plant material was separated into root system, stems, leaves, flowers and fruits, which were washed with deionized water, oven-dried and homogenized. The plant materials were digested by Multiwave Go Plus microwave digestion system (Anton Paar) with $HNO_3$ and analysed for the determination of As, Cd, Cr, Cu, Ni, Mn, Pb, Zn, Fe, Hg, P concentrations by inductively coupled plasma optical emission spectrometry (ICP-OES) method as well as inductively coupled plasma mass spectrometry (ICP-MS). The ICP-OES analysis was performed using an ICP-OES 5100SVDV spectrometer (Agilent) in axial view. The ICP-MS analysis was performed using an ICP MS 7900 spectrometer (Agilent) in both gas-free and He-mode.

Physicochemical characterization of soil samples after the experiment and substrate was carried out to determine pH, EC and carbonate content. The methodology is described

above. Microbial analysis of soil samples after maize harvesting was performed using ARISA (Automated ribosomal intergenic spacer analysis), in which total metagenomic DNA (mgDNA) is isolated from control soil substrate as well as soil substrates with 1% addition of SS and BC after the end of the maize sowing experiment. For analysis, DNA was eluted in 50 µL of sterile water and then the concentration and purity of the DNA samples were measured spectrophotometrically using a NanoDrop One instrument (Thermo Fisher Scientific). Later, the samples were diluted to a uniform concentration of 25 ng/µl and stored at $-20$ °C. ITSF and ITSReub primers (*Cardinale et al., 2004*) were used for PCR (polymerase chain reaction) amplification of the bacterial 16S-23S rRNA variable ITS (intergenic transcribed spacer) region, and the reverse primer was fluorescently labeled at the 5′-end with 6-FAM. The length-variable DNA region of fungal ITS1—5.8S rRNA—ITS2 was amplified using the 2234C and 3126T primer pairs (*Sequerra et al., 1997*), and in this case the forward primer was fluorescently labelled at the 5′-end using 6-FAM. PCR amplification was performed in a GeneAmp PCR System 9700 (Applied Biosystems) under the following conditions: initial denaturation for 3 min at 94 °C; followed by 35 cycles of denaturation for 45 s at 94 °C, annealing of the primers for 1 min at 60 °C, and polymerization for 2 min at 72 °C; and final polymerization for 10 min at 72 °C. The PCR products were subsequently precipitated with ethanol and dissolved in 10 µL of sterile water. To 1 µL of purified PCR product was added 9 µL of formamide containing LIZ1200 size standard (Applied Biosystems), the mixture was denatured for 3 min at 95 °C and separated by capillary electrophoresis in an ABI 3100 Prism Avant instrument (Applied Biosystems). Data obtained from fragment analysis with a fluorescence value greater than 30 were analysed using Peak Scanner 2 (Applied Biosystems) software, and PCR product sizes ranging from 200 to 1,000 bp were used for statistical evaluation.

## Experimental design and statistical analysis

All physicochemical characterizations were performed in a minimum of three or four repetitions. In the case of elemental analysis, this involved the repeated determination of a single representative sample with a standard error of measurement not exceeding 5%. The data obtained were subjected to basic statistical processing to obtain the mean and standard deviation values. For statistical assessment and graphical processing of data from characterization and cultivation experiment the ANOVA with Tukey post-hoc test (significance level $\alpha = 0.05$) has been used. All analyses were performed with SPSS *v.* 26 (IBM Corp., Armonk, NY, USA) and OriginPro 2016 (OriginLab Corporation, Northampton, MA, USA). For microbial analysis, Venn diagrams were created using an online tool at: https://bioinformatics.psb.ugent.be/webtools/Venn/. Standardized skewness and standardized kurtosis were used to determine the normal distribution of the data. As their values were not outside the expected range, statistically significant differences between variants were assessed using ANOVA at the 95% significance level followed by a post hoc LSD test for pairwise comparisons with a 95% confidence limit. Simpson, Shannon, Evenness and Chao-1 indices were used to detect alpha diversity of the microbial community using the PAleontological STatistics (PAST) software version 3.19. This statistical software was also used for principal component analysis (PCA). Principal

component (PC) scores with a variability of over 1% from the PCA analysis were used for ANOVA analysis and to detect statistically significant differences in microbial diversity among treatments in the multivariate PCA analysis.

## RESULTS

### Physico-chemical characteristics

Based on the determination of the potential pH of the BC and SS samples (Table 1), it can be concluded that the pyrolysis treatment of the distillery sludge at 400 °C increases the pH of the material. The electrical conductivity (EC) is a parameter that characterizes the representation of the total amount of water-soluble salts in the studied materials. In our case, the pyrolysis treatment of the distillery's sewage sludge at 400 °C resulted in a significant reduction of the EC value by more than 50% (Table 1), indicating that the representation of water-soluble salts in the BC was reduced. In the SS sample as input material, the ash content was 23.59% and after pyrolysis treatment at 400 °C, the ash content in the biochar obtained increased to 50.99%. In the analysis of our samples, we found that pyrolysis treatment of sewage sludge at 400 °C resulted in approximately twofold decrease in CEC in BC (55.656 cmol/kg) compared to SS (118.66 cmol/kg) (Table 1). Pyrolysis treatment of the centrifuged sewage sludge enriched the obtained BC solid material in total C, with a significant increase in the $C_{org}$ (organic carbon) form (36, 83% for BC and 28.86% for SS) and a decrease in the $C_{inorg}$ (inorganic carbon) form (1.73% for BC and 2.65% for SS). The pyrolysis treatment of the sludge reduced the % of bound H in the obtained biochar (Table 1), and the H/C ratio also decreased significantly (from 0.234 for SS to 0.051 for BC). After pyrolysis treatment, the C/N ratio also increased in BC compared to SS. RFA analysis of both samples determined the highest concentration of Fe (135,000 mg/kg for BC and 107,000 mg/kg for SS) and Zn (1,656 mg/kg for BC and 593 mg/kg for SS). Other identified heavy metals and potentially toxic elements were represented in the samples in the following order in terms of decreasing concentration Cu >Ni >Cr >Pb >Hg >Cd >As. The RFA analysis values obtained for the total concentration of metals and metalloids present in BC and SS (Table 1) indicate that pyrolysis treatment at 400 °C results in approximately a twofold increase in Cr, Pb, Cu and a threefold increase in Ni and Zn in BC. For BC and SS samples, the concentrations of As and Cd were below the detection limit of the analytical method (0.1 mg/kg). The amount of Hg in the BC sample was reduced by pyrolysis treatment to below the detection limit (0.2 mg/kg).

The total concentration of polycyclic aromatic hydrocarbons (PAHs) present was reduced by pyrolysis treatment of the industrial sewage sludge to 3.38 mg/kg in the final BC material from 4.03 mg/kg in SS (Table 2). In general, the concentration of the individual PAHs of the structures was reduced by pyrolysis treatment at 400 °C, with naphthalene being the most abundant in both materials (1.82 mg/kg for SS and 1.68 mg/kg for BC). In the toluene extracts of the analysed materials, other 4-ring and 3-ring PAHs structures were also identified above the detection limit of the assay, as well as the structure of a single 5-ring PAH, namely benz(b)fluoranthene. Benzo(b)fluoranthene was also the only PAH structure detected whose total extractable concentration was increased by pyrolysis treatment in the BC compared to the input precursor SS.

**Table 1** Physico-chemical characterisation of sewage sludge (SS) and sewage sludge-derived biochar. (BC).

|  | BC | SS |
|---|---|---|
| pH | $7.71 \pm 0.02$ | $7.33 \pm 0.01$ |
| EC (mS/cm) | $1.06 \pm 0.01$ | $2.35 \pm 0.01$ |
| CEC (cmol/kg) | $55.66 \pm 2.67$ | $118.66 \pm 3.41$ |
| $C_{total}$ (%) | $38.56 \pm 0.06$ | $31,51 \pm 0.06$ |
| $C_{org.}$ (%) | $36.83 \pm 0.06$ | $28.86 \pm 0.06$ |
| $C_{inorg.}$ (%) | $1.73 \pm 0.06$ | $2.65 \pm 0.06$ |
| N (%) | $6.53 \pm 0.03$ | $6.19 \pm 0.02$ |
| H (%) | $1.89 \pm 0.02$ | $6.76 \pm 0.05$ |
| $H/C_{org}$ | 0.05 | 0.23 |
| $C_{org}/N$ | 5.64 | 4.66 |
| ash content (%) | $50.99 \pm 0.03$ | $23.59 \pm 0.03$ |
| As (mg/kg) | <1 | <1 |
| Cd (mg/kg) | <1 | <1 |
| Cr (mg/kg) | 93 | 42 |
| Cu (mg/kg) | 295 | 158 |
| Ni (mg/kg) | 151 | 51 |
| Pb (mg/kg) | 32 | 15 |
| Zn (mg/kg) | 1656 | 593 |
| Fe (mg/kg) | 135000 | 107000 |
| Hg (mg/kg) | <2 | 2 |

## Short-term ecotoxicological tests

Graphical representations of the results of the three-stage ecotoxicological germination test are shown in Fig. 1. In all three levels, *i.e.,* when considering I. the influence of volatile contaminants, II. the influence of volatile contaminants, water-soluble contaminants and direct contact with BC and SS, and III. the influence of volatile and water-soluble contaminants, we did not observe a statistically significant difference between the additions ($\alpha = 0.05$). Comparing the level II control group and with the group with 1% addition of BC, we found that seed germination did not increase statistically significantly with the addition of biochar. Even, substrate enrichment with 1% addition of BC inhibited germination most at level II, when seeds are in direct contact with biochar while being exposed to volatile and water-soluble contaminants from BC. However, even in this case, no statistically significant difference was confirmed at the $\alpha = 0.05$ level of statistical significance. In the case of 1% addition of untreated SS, seed germination is most inhibited at level III, where seeds are exposed to volatile and water-soluble contaminants.

We also assessed the ecotoxicity of BC and SS in terms of their impact on soil biota by conducting a 4-month incubation test with soil giant nematodes, specifically California earthworms (*Eisenia fetida*). During the exposure period, 120 days, we monitored earthworms every 30 days, and mortality of 10 individuals in all groups began to be observed after three months. The mortality of earthworm individuals after 90 days of exposure was 30% in the control sample (without BC and SS addition), 60% in the sample

**Table 2  Concentration of 18 PAHs in SS and BC samples.**

| PAH structure | SS (mg/kg) | BC (mg/kg) |
|---|---|---|
| naphthalene | $1.82 \pm 0.01$ | $1.68 \pm 0.01$ |
| acenaphthylene | –[*] | –[*] |
| 1-methylnaphthalene | $0.28 \pm 0.01$ | $0.27 \pm 0.01$ |
| 2-methylnaphthalene | $0.79 \pm 0.01$ | $0.64 \pm 0.01$ |
| acenaphthalene | –[*] | –[*] |
| fluorene | –[*] | –[*] |
| phenanthrene | $0.26 \pm 0.01$ | $0.16 \pm 0.01$ |
| anthracene | –[*] | –[*] |
| fluoranthene | –[*] | –[*] |
| pyrene | $0.16 \pm 0.01$ | –[*] |
| benzo(a)anthracene | $0.21 \pm 0.01$ | $0.10 \pm 0.00$ |
| chrysene | $0.49 \pm 0.01$ | $0.26 \pm 0.01$ |
| benzo(b)fluoranthene | $0.01 \pm 0.00$ | $0.27 \pm 0.01$ |
| benzo(k)fluoranthene | –[*] | –[*] |
| benzo(a)pyrene | –[*] | –[*] |
| dibenzo(a,h)anthracene | –[*] | –[*] |
| benzo(g,h,i)perylene | –[*] | –[*] |
| ideno(1,2,3-c,d)pyrene | –[*] | –[*] |
| Σ 18 PAHs | 4.03 | 3.38 |

**Notes.**
[*]Concentrations under LOD of analytical determination (0.10 mg/kg).

with 1% SS addition, and 20% in the sample with 1% BC addition. We did not observe cocoon formation in either sample for the duration of the experiment. After the end of the experiment, an exposure period of 120 days, there were seven original individuals in the control sample, two individuals in the substrate with 1% SS addition and four individuals in the substrate with 1% BC addition. Earthworm mortality decreases in the order SS >BC >CONTROL, while the average weight of surviving individuals increases in the reverse order. The total concentration of selected heavy metals (Fe, Zn, Cu, Pb and Cd) in the earthworm digests obtained from control, 1% SS and 1% BC additions were determined using both F-AAS and ET-AAS, and all the values obtained did not exceed 1 mg/kg. The Fe value in individuals from the 1% SS exposure was almost threefold higher compared to individuals from the 1% BC exposure, and almost sixfold higher compared to the control group. Similarly, the amount of Cu and Cd was lower in individuals from the 1% BC exposure compared to individuals from the 1% SS exposure. There was also no statistically significant difference between the amount of Zn in individuals from the 1% BC exposure compared to individuals from the 1% SS exposure. In general, total heavy metal levels decreased in the control and 1% BC groups in the order Zn >Fe >Cu >Pb >Cd and in the 1% SS group in the order Fe >Zn >Cu >Pb >Cd.

## Pot experiment with *Zea mays* L.

Throughout the outdoor experiment, we recorded the height of the plants in the pots every 7 days, based on which we established a trend of plant growth. From the data obtained,

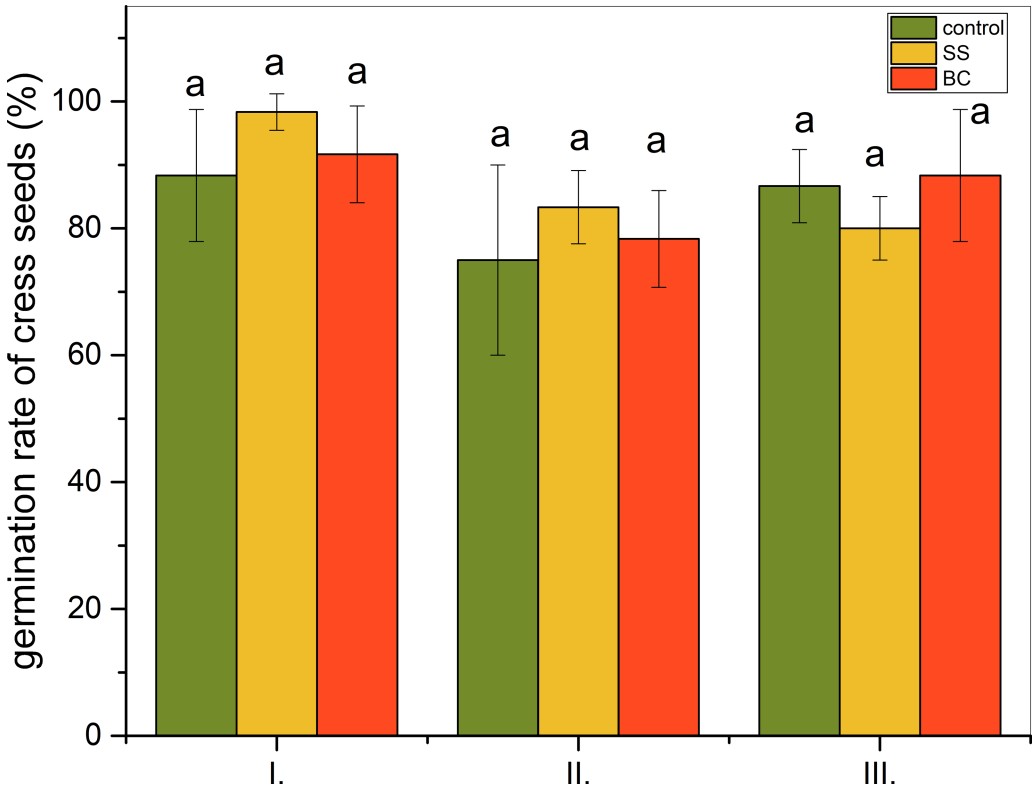

**Figure 1 Ecotoxicological germination test (%) of cress seeds (*Lepidivum sativum* L.).** I. Effect of volatile contaminants, II. effect of volatile contaminants + effect of water soluble contaminants + direct contact with BC and SS, III. effect of volatile and water soluble contaminants. Control = sand without addition of SS and BC. Different letters under impacts I., II. and III. indicate a statistically significant difference (Tukey *post-hoc* test, $\alpha = 0.05$, degrees of freedom = 4), statistical difference between impacts I., II., and III. was not assessed.

we conclude that the growth of the above-ground part of the maize was slower in all pots for the first 30 days. After one month, the growth rate of the plants increased, and we observed the greatest increase 60-80 days after planting. After about 80 days from planting, we observed a decreasing trend in plant growth. From the data obtained, it is evident that the plant height was highest in the substrate enriched with 1% addition of SS followed by plants with 1% addition of BC and the lowest plants were in the control group. The average values of the dimensions (height and length, respectively) and weight after harvesting (22 weeks from sowing) and drying of the underground and aboveground parts of the plants are given in Fig. 2. From the graph (Fig. 2A), it can be seen that the weight of the above and below ground parts of the control group plants is almost the same, while the height of the above ground part is three times higher. In the case of the plants enriched with 1% SS addition, it can be seen from the graph that the difference in the weight of the biomass of the below ground part is statistically significant compared to the weight of the biomass of the below ground part of the plants grown in the substrate with 1% BC addition. In the case of comparison of above-ground part biomass weight between plants grown in 1% SS substrate
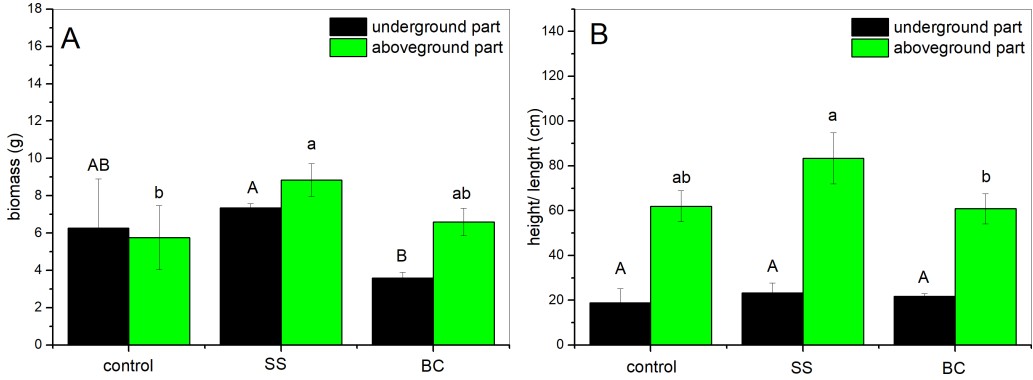

**Figure 2** Biomass weight (A) and average height and length (B) of aboveground (green bars) and underground (black bars) parts of a corn after harvesting ($n = 3$). Different letters indicate a statistically significant difference between additions (Tukey *post-hoc* test, $\alpha = 0.05$, degrees of freedom $= 4$).

and 1% BC substrate, statistical significance was not confirmed ($\alpha = 0.05$). We did not observe statistically significant differences between the biomass of the underground part and the aboveground part of SS and control. The highest plant biomass yield was obtained from plants from the group with 1% SS addition. Comparison of the height and length of above-ground plant individuals grown in the control (no addition), 1% SS and 1% BC addition environments (Fig. 2B) showed a statistically significant difference between plants grown in the SS and BC environments, respectively (Fig. 2B). However, both treatments did not show statistically significant difference with control environment. The lengths of the underground parts (root systems) were without statistically significant difference among all groups analyzed.

Based on the experimental data obtained, we found that the soil substrate used has a pH in the acidic region (6.4), and after cultivation of maize, the pH increased in all soils, in the order of CONTROL <SS <BC. The most significant increase in soil pH occurred in the case of 1% addition of BC to the soil (7.21). In both groups (BC, SS), we observed an increase in pH and a decrease in $C_{anorg}$, with the largest increase and decrease occurring simultaneously in the case of soil with 1% BC addition. In the case of the group with 1% addition of SS, we observed an increase in the amount of $C_{anorg}$ in the soil after the end of the experiment, which may be due to the higher amount of $C_{anorg}$ in SS compared to BC. In terms of soil EC, we found that after cultivation of maize, EC decreased in the order CONTROL >SS >BC.

Total microbial biomass activity was assessed as the amount of DNA isolated per 1 g of soil substrate. The DNA isolated represents DNA from both living and dead microorganisms. Using the applied least significant difference (LSD) statistical test, there was a statistically significant difference between the total microbial biomass in the BC amended soil and the control (LSD, $\alpha = 0.05$), with the BC amended soil showing the statistically lowest microbial biomass (Fig. 3). The representation of microbial biomass in the soil substrates decreased in the order CONTROL >SS >BC. Venn diagrams were used to evaluate the common and dissimilar bacterial (B-ARISA) and fungal (F-ARISA) species among the samples (Fig. 4).

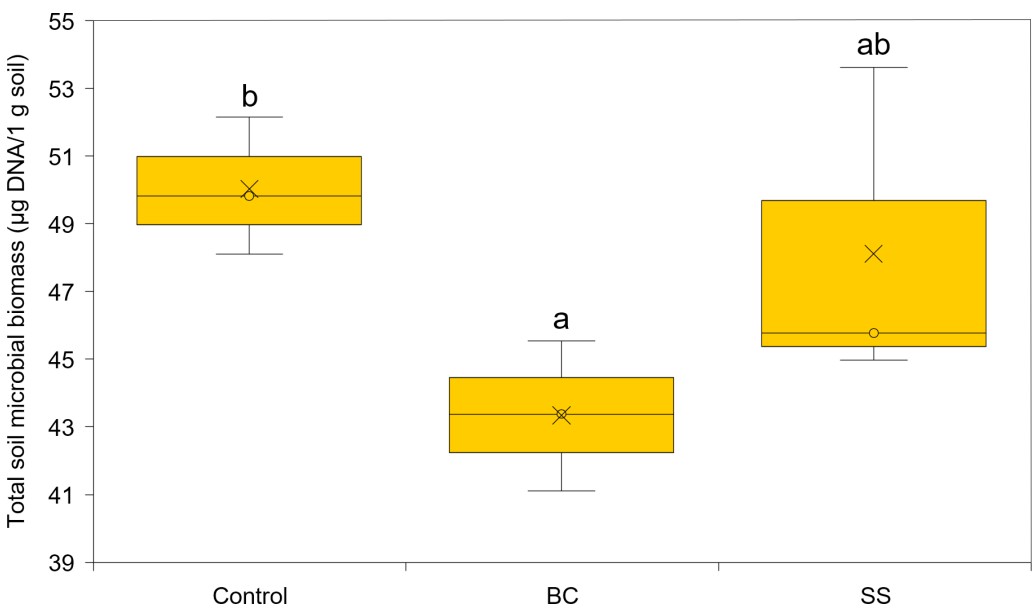

**Figure 3 Total microbial biomass expressed as the amount of DNA per 1 g of soil substrate.** The cross represents the mean and the circle the median, and the letters a, b, and ab represent statistically significant difference between treatments (LSD, $\alpha = 0.05$, degrees of freedom = 4).

Soil with 1% BC addition contained the lowest number of bacterial species but on the other hand the highest number of fungal species. The number of bacterial species decreased in the order CONTROL >SS >BC, with 44 fewer bacterial species in the soil with 1% BC compared to the soil with 1% SS addition and up to 50 fewer species in the soil with 1% BC addition compared to the control. For fungi, this variation in species numbers was not as pronounced and the soil enriched with 1% BC had six more fungal species than the other samples. Of all the bacterial and fungal species detected, eight bacterial species and 15 fungal species were identified in all three samples.

From our results (Fig. 5A), it is evident that maize is a Zn hyperaccumulator as we identified increased translocation of Zn to above ground parts in all samples. Zn accumulation by maize roots was statistically significantly reduced after 1% addition of SS and BC compared to the control. For SS plants, although there was a statistically significant decrease in the amount of Zn in the stem of maize compared to the control, based on % translocation, we found that the largest amount of Zn accumulated in the aboveground part was in SS plants. The translocation of Fe uptake in the maize specimens obtained and Fe accumulation in the different plant parts is shown in Fig. 5B, and it can be observed that the difference between the groups analysed is not statistically significant in any plant part. In all cases, individuals accumulated more Fe in the roots than in the above-ground part, with the % Fe abundance in the different parts decreasing in the order R (root) >L (leaf) >F (flower) ≥ S (stem) >C (cob-fruit). The translocation of As uptake in the maize specimens obtained and the accumulation of As in the different plant parts is shown in Fig. 5C, and it can be observed that the difference between the analysed groups is not

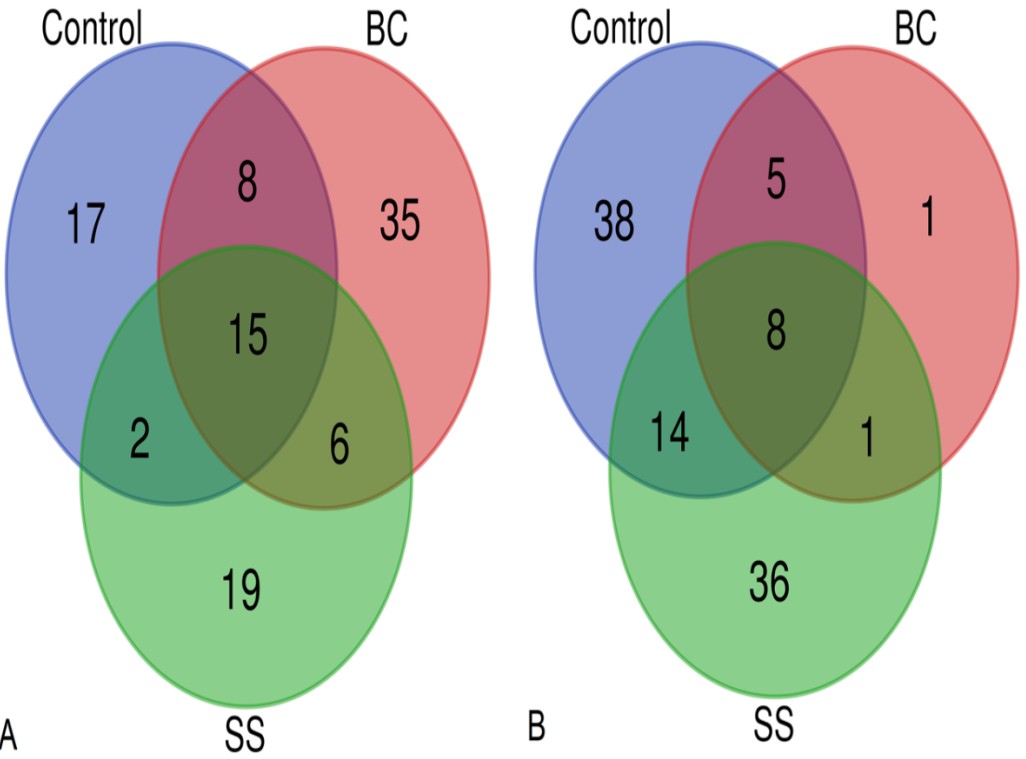

**Figure 4** Venn diagrams showing the number of common and different fungal species (A) and common and different bacterial species (B) between control soil, soil with 1% BC addition and soil with 1% SS addition.

statistically significant in any plant part. For all groups analysed, the amount of As in stems, flowers and fruits was lower than the limit of detection (LOD) of the analytical method of determination used (0.4 mg/kg). Individuals from the control and SS groups accumulated more As in the roots than in the aboveground part, and in the case of BC individuals we observed an increase in translocation to the aboveground part, resulting in the same percentages of As in roots and leaves. For Cd, similar to As, we did not expect a statistically significant effect of 1% addition on Cd accumulation and translocation in maize, since both the soil substrate and the additions contained less than 1 mg/kg Cd according to the RFA analysis. The translocation of ingested Cd in the obtained maize individuals together with the accumulation of Cd in the different plant parts is shown in Fig. 5D, and it can be observed that the differences between the amount of Cd in the root, stem and leaves are not statistically significant between the analysed groups. Compared to the other plant parts, the highest amount of Cd was accumulated in roots within all groups. A statistically significant reduction in Cd content was observed only in the BC maize flower compared to the control. Also, for BC cobs we observe a reduction in Cd content in fruits and also a reduction in % translocation of Cd to fruits. The translocation of the Cr uptake in the maize specimens obtained and the accumulation in the different plant parts is shown in Fig. 5E, and it can be observed that the difference between the analysed groups is not statistically significant in

any plant part. From the results obtained, it can be assumed that in the soil substrate itself, Cr was mainly represented in the form of Cr(VI), since in the case of the control, 61% of the Cr taken up was transported to the aboveground part. After application of 1% SS and 1% BC, the amount of accumulated Cr in the roots increased statistically insignificantly. For all samples, the highest amount of Cu accumulated was in the roots, with no statistically significant difference between the groups. However, in SS plants we observed that up to 68% of Cu was accumulated in the whole aerial part (Fig. 5F). From the results obtained, it is evident that 1% addition of SS and BC did not have statistically significant effect on the amount of Cu accumulated by plant parts compared to control, but in case of 1% addition of BC, there was a change in Cu translocation in maize plant parts compared to control and SS. We observe a statistically significant increase in the amount of Ni in the stem in BC plants compared to SS and a statistically significant decrease in the amount of Ni in the flowers in SS compared to control. Thus, Ni accumulates in SS and BCS plants mainly in the non-edible parts of the plants, which protects the fruits and seeds from contamination. In the case of BC maize, we observe an overall increase in the amount of Ni uptake into the aboveground part of the maize (Fig. 5G). For our samples, we identified (Fig. 5H) that it was the roots and leaves that accumulated the most Pb, while in the case of roots we did not find a statistically significant difference between the analyzed groups and, on the contrary, in the case of leaves we identified a statistically significant decrease in the amount of Pb in SS plants compared to the control. In the case of maize plant stems, we cannot assess whether there was a statistically significant difference in accumulation, as the amount of Pb in SS plants was below the detection limit of the assay analytical method used (0.1 mg/kg). The translocation of uptake of P in the maize specimens obtained and the accumulation in the different plant parts is shown in Fig. 5I, and we found that there were no statistically significant changes in the roots, leaves, flowers and fruits due to the effect of 1% addition of BC or SS compared to the control. However, in the case of SS plants, there was a statistically significant decrease in the amount of P accumulated in the stem, which is probably the result of increased translocation of P from the stems of SS plants to the fruits, in order to maintain an optimal level of P in the leaves, consumed for photosynthesis. The translocation of uptaken Mn in the obtained maize individuals and the accumulation in the different plant parts is shown in Fig. 5J, and it can be observed that the 1% addition of BC and neither SS statistically significantly affects the amount of Mn in the stems and roots. However, 1% addition of BC statistically significantly reduced the amount of Mn in leaves compared to the control and also along with 1% addition of SS statistically significantly reduced the amount of Mn in flowers compared to the control.

## DISCUSSION

The obtained pH value of the untreated sludge (7.33) is comparable to the average pH value of sludge from industrial wastewater treatment plants analysed in the work of *Kazi et al. (2005)*. Only a very small increase in the pH value was observed when the sewage sludge was thermochemically treated at a relatively low temperature of 400 °C, which was comparable to the pH increase in the work of *Hossain et al. (2011)* for pyrolysis treatment of sewage sludge at an analogous temperature. The increase in pH during the

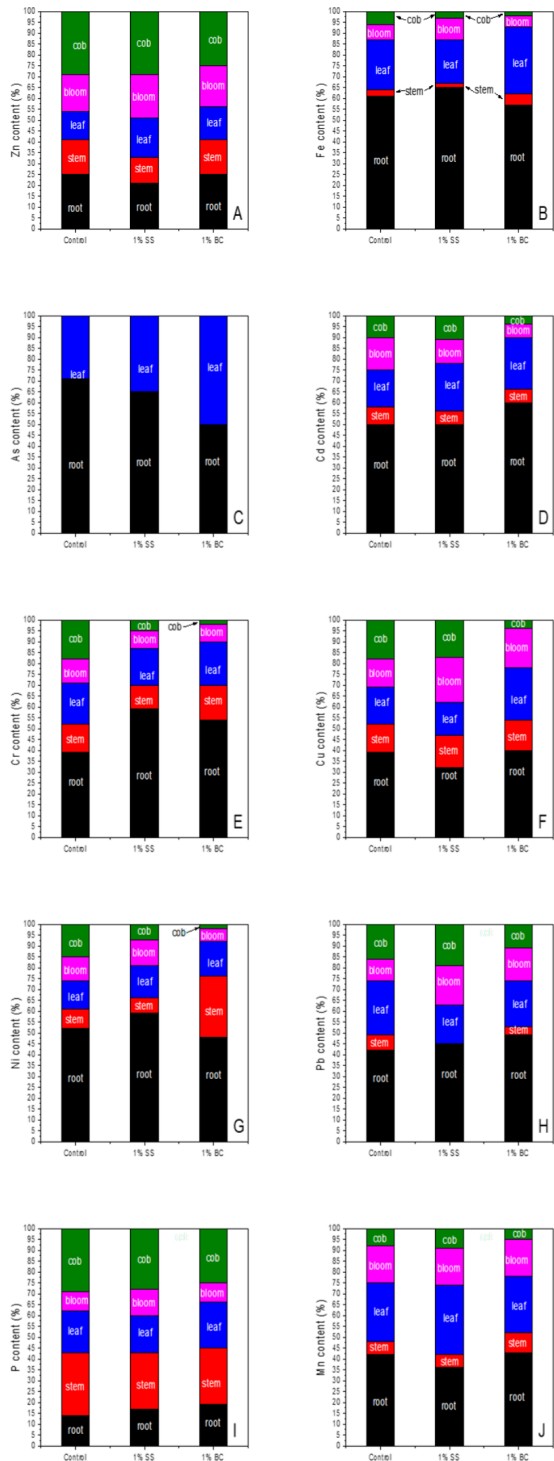

**Figure 5** Translocation of the studied elements Zn (A), Fe (B), As (C), Cd (D), Cr (E), Cu (F), Ni (G), Pb (H), P (I), Mn (J) in plant parts of maize cultivated in the medium of 1% addition of BC, SS and control medium (without addition).

pyrolysis treatment of sludge, according to *Souza et al. (2021)* is due to a decrease in the concentration of functional groups present (*e.g.*, carboxyl and hydroxyl), which are acidic in nature, while the alkalinity of the resulting biochar increases with increasing pyrolysis temperature. Obtained EC values of SS and BC were similar to values found by *Souza et al. (2021)*, according to whom the EC value itself is closely related to the ash content of the analysed material. During the pyrolysis of sludge, the ash content increases, while the solubility of salts and metals in water decreases, and thus the EC value of the material decreases. The authors attribute the decrease in solubility of salts and metal compounds at pyrolysis temperatures above 200 °C to the fixation of $K^+$, $Ca^{2+}$, $Mg^{2+}$ and $PO_4^{3-}$ ions in the mineral fractions, while the mineral content in the analysed material is directly related to the ash content. Sewage sludge contains, according to *Wentzel et al. (2002)* generally around 25% ash. In the SS sample, the ash content was 23.59% and after pyrolysis treatment at 400 °C the ash content of the obtained biochar increased to 50.99%. The CEC value of biochar depends mainly on the nature of the biomass feedstock and the pyrolysis temperature. Similar results to those obtained in our study for SS and BC samples were also obtained by *Figueiredo et al. (2021)* according to which the decrease in the CEC of sewage sludge-based biochar prepared at temperatures above 200 °C is due to the elimination of oxygen functional groups on the surface of the produced biochar. The results obtained for C, H, N contents in SS and BC materials correspond with the results of the work of *Fachini & Figueiredo (2022)*, who justify the increase of total C in sewage sludge-based biochar by the formation of more stable organic compounds in the resulting solid pyrolysis product due to the reorganization of aliphatic hydrocarbon compounds into aromatic compounds. The H/C ratio is used to indirectly indicate the degree of aromaticity of the resulting biochar, and according to *Yuan et al. (2013)* the lower the value of this ratio the higher the number of aromatic structures in the obtained material. An increase of N in biochar obtained at pyrolysis temperatures up to 400 °C was also found by *Figueiredo et al. (2017)*. The authors justify this fact by the presence of more thermally stable N compounds in the sludge, which do not decompose up to 400 °C. The C/N ratio of BC also increased after pyrolysis treatment compared to SS, which corresponds with the results of *Figueiredo et al. (2017)*, according to which the increase of C/N in sewage sludge-derived biochar is due to the increase of $C_{org}$ in the biochar and the transformation of N into gaseous form. The C/N ratio is used as an indicator of the ability of organic matter to release nitrogen when biochar is used as a soil additive (*Jin et al., 2016*).

The total concentrations of extractable PAHs found under the given experimental conditions in both analysed materials meet the limit (6 mg/kg) set for soil additives under the EU Fertiliser Regulation (*Regulation 2019/1009 of the European Parliament and of the Council, 2019*). Our results correspond with the results of *Buss et al. (2022)*, who found naphthalene to be the dominant PAHs structure (72 ± 21%) in biochar produced from sewage sludge and lignocellulosic biomass. HPLC analysis of toluene extracts of BC and SS aimed at quantification of methylated hydrocarbons revealed an equally significant representation of 1-methylnaphthalene and 2-methylnaphthalene structures. The methylated hydrocarbons have higher volatility than other PAHs structures, which makes them more dangerous for the environment. The presence of PAHs in BC poses a

potential ecotoxicological risk when this material is applied as a soil additive. However, the concentrations of PAHs structures determined by us represent the total, extractable concentrations present in BC and SS under given conditions, and thus are not readily releasable and bioavailable concentrations of PAHs.

According to *Chandra & Kumar (2016)*, the elevated concentration of heavy metals in sludge from distillery WWTPs is due to the corrosive effect of sugarcane juice components as well as the potential addition of metal-containing additives during fermentation and distillation processes. The increase of heavy metals in biochar occurs due to higher boiling temperatures of heavy metals, resulting in a higher mass loss of organic compounds than the mass loss of heavy metals during pyrolysis. Increases in Cd, Cr, Cu, Ni, Pb and Zn after pyrolysis of sewage sludge were also observed by *Li et al. (2021)*. Based on the findings, we conclude that the content of heavy metals and potentially toxic elements in BC is directly related to their content and form in the input precursor, sewage sludge. The obtained values of heavy metals in BC were compared with the limits determined by EBC (European Biochar Certificate) standards for biochar of EBC classification class—Agro. Based on these limits, the BC sample meets the permissible concentration for As, Cd, Pb. The Cu and Ni contents are three times higher, while the Zn content is almost four times higher than the *EBC (2022)* limits. According to the findings of *He et al. (2010)*, pyrolysis temperatures above 350 °C leads to higher concentrations of Cd, Pb, Zn, and Cu in sewage sludge-based biochar, but at the same time there is a stabilization in the carbon structure, which reduces their bioavailability to plants. The values obtained for the total concentrations of heavy metals and potentially toxic elements present in the input precursor SS were compared with Annex II of *Directive 86/278/EEC (1986)*, which provides limit values for the concentration of heavy metals in sludges used as soil additives. We found that the heavy metal content of SS complies with the values set out in the Directive, on the basis of which we conclude that untreated SS can be used as a soil additive in terms of heavy metal content. From the point of view of meeting the BC limits, it is inevitable to consider the pre-treatment of the input sludge or the preparation of a mixed precursor based on sludge and other material, which would reduce the overall concentration of elements in the final product.

By RFA analysis, we equally quantified the abundance of heavy metals and potentially toxic elements in the soil substrate used in the outdoor pot experiment under realistic conditions with a maize (*Zea mays* L.) . We found that in the soil substrate the elements are represented in the following order in terms of decreasing concentration Fe >Cu >Cr >Zn >Ni >Pb >As >Hg >Cd, with all the values of heavy metals and potentially toxic elements present meeting the limit values for heavy metal concentrations in soil with a pH in the range of 6-7 according to Annex I of *Directive 86/278/EEC (1986)*.

Germination tests confirmed that neither BC nor SS had a negative effect on cress seed germination compared to the control at any of the three levels. According to *Bouqbis et al. (2016)* biochar increases the water holding capacity of the soil thereby positively affecting seed germination. The same results of the three-level ecotoxicological test were obtained by *Buss & Mašek (2014)*, who investigated the effect of 1%, 2% and 5% biochar addition on seed germination of cress (*Lepidium sativum*) seeds.

The accumulation of heavy metals in earthworm tissues occurs through two mechanisms, namely chemical sorption of soluble forms of metals through the epidermis and digestion (*Malińska et al., 2017*). The higher weight of individuals in the 1% SS supplemented sample may be due to the higher number of bioavailable forms of heavy metals and nutritionally important elements in SS compared to BC, which accumulate in earthworm tissues, increasing their weight. The increase in earthworm weight in the exposure with 1% addition of SS corresponds with the work of *Kniuipyté et al. (2020)*, who found that low addition of sewage sludge to soil increases earthworm weight as sludge is a source of nutritionally important substances that promote their growth. In general, total heavy metal levels decreased in the control and 1% BC groups in the order Zn >Fe >Cu >Pb >Cd and in the 1% SS group in the order Fe >Zn >Cu >Pb >Cd. Since the experimental data obtained, we can conclude that the pyrolysis treatment of the sewage sludge from the distillery's WWTP, at a temperature of 400 °C, resulted in a multiplication of Fe, Cu, Pb and probably also Cd in the resulting pyrolysis product, but on the other hand, the bioavailable forms of these metals for living organisms were reduced.

The declining trend in plant growth, according to *Rout & Sahoo (2015)*, is due to leaf senescence and decrease in leaf area, resulting in reduced assimilation of light energy required for photosynthesis, was observed in our study. From the data obtained, it is evident that the plant height was highest in the substrate enriched with 1% addition of SS followed by plants with 1% addition of BC and the lowest plants were in the control group. According to *Ason et al. (2015)*, root weight is positively correlated with the above ground height of the plants as the root system absorbs water and nutritionally important substances for the stem and leaves, which in turn produce nutrients to maintain the root system. This statement can be confirmed especially in the case of plants enriched with 1% SS addition, it is clear from the graph (Fig. 2A) that the difference in the weight of the biomass of the under-ground part is statistically significant compared to the weight of the biomass of the above-ground part of the plants grown in the substrate with 1% BC addition. The reduction in root system weight due to 1% BC can be attributed to the reduction in root system weight according to *Prendergast-Miller, Duvall & Sohi (2013)* due to fewer and especially thinner root hairs through which the plant absorbs nutrients. The highest plant biomass yield was obtained from the plants of the group with 1% SS addition, which is the highest yield according to *Khanmohammadi, Afyuni & Mosaddeghi (2016)* due to faster release of bioavailable N from untreated sewage sludge as opposed to biochar from sewage sludge. The most significant increase in soil pH occurred with 1% BC addition to the soil (7.21), which corresponds with the results of the work of *Junior & Guo (2023)*, since biochar generally tends to reduce soil acidity. According to *Wang, Felice & Scow (2020)*, growing maize reduces the amount of $C_{inorg}$ in the soil, and the reduction in $C_{inorg}$ is directly related to the increase in soil pH, as the released $Ca^{2+}$ reacts with $OH^-$ to form alkaline hydroxide. This statement corresponds with the results obtained for the soil from the control group and from the group with 1% BC addition. In both groups, we observed an increase in pH and a decrease in $C_{inorg}$, with the greatest increase and decrease occurring in the case of soil with 1% BC addition. In the case of the group with 1% addition of SS, we observed an increase in the amount of $C_{inorg}$ in the soil after the end of the experiment,

which may be due to the higher amount of $C_{inorg}$ in SS compared to BC. The claim by *Wu et al. (2010)* also corresponds with our findings regarding plant biomass yield, as we observed the highest above-ground plant biomass yield for plants from the 1% SS addition group (Fig. 2A). In terms of soil EC, we found that EC decreases after cultivation of maize, which according to *Zhang et al. (2015)* due to the growth of microorganisms in the soil. The EC decreased in the order of CONTROL >SS >BC. Based on the results obtained, we hypothesized that the more the EC was reduced the greater the amount of microorganisms would be found in the soil. We verified our assumption by microbiological analysis of soil samples after the experiment.

Changes in microbial diversity of soil amended with SS and BC can be discussed in several ways. According to *Wang, Felice & Scow (2020)*, a reduction in total microbial biomass due to biochar addition can occur if the biochar contains significant amounts of toxic PAH structures and volatile organic compounds (VOCs). As was discussed above, after pyrolysis treatment of sludge, the resulting biochar was enriched in the PAH structure benzo(b)fluoranthene, which according to *Yang et al. (2022)* has a toxic effect on soil bacteria. A decrease in microbial biomass due to the addition of biochar to sandy soil was also observed by *Brtnicky et al. (2019)*. According to the authors, the decrease in microbial biomass may be due to adsorption of nutrients from the soil onto the biochar surface, resulting in reduced nutrient availability to microorganisms. The representation of microbial biomass in soil substrates decreased in the order CONTROL >SS >BC, and the obtained result does not confirm our established assumption regarding the decrease of soil EC due to the increase of microorganisms in the soil. The result obtained corresponds with *Yao et al. (2017)*, who describe that the increase in fungal species variation in biochar-enriched soil can occur due to the improvement in aeration and water-holding capacity of the soil, and these changes favourably affect the habitat of soil fungi and the development of fungal filaments, hyphae. According to *Zheng et al. (2016)*, the increase in fungal species variation by the addition of biochar may also occur due to a decrease in soil bacteria, specifically streptomyces, which produce antifungal compounds. Higher amounts of fungi in the soil, especially mycorrhizal fungi, can reduce the root-shoot translocation factor of heavy metals in the plant, due to the immobilization of metals by hyphae formed by the fungus in the soil and on the root surface (*Gómez-Gallego et al., 2022*). Based on this, we hypothesize that in plants from soil enriched with 1% BC addition, a smaller amount of heavy metals will be accumulated in the aboveground part and a larger amount will be accumulated in the root. According to *Jamal et al. (2002)*, higher number of mycorrhizal fungi in the soil causes a reduction in root weight and branching in cereals, as the fungal hyphae increase the absorption area of nutrients from the soil instead of the root. This statement corresponds with our results, as we found a statistically significant decrease in root weight in BC soil plants, which may be due to an increase in fungal species variability, specifically the occurrence of mycorrhizal fungi in the soil substrate enriched with 1% BC addition. The results of the Venn diagrams (Fig. 4) also indicate that there are no statistically significant changes in the species variation of bacteria and fungi in the soil with 1% SS addition compared to the control, which is in agreement with the results of *Hawrot-Paw et al. (2022)*. However, *Hawrot-Paw et al. (2022)* found that although there is no increase
in the species variability of bacteria and fungi in the soil during maize cultivation due to the impact of the sewage sludge, there is a change in the abundance of bacteria and fungi in the soil. They identified a higher abundance of *Bacillus* bacteria, which stimulate maize growth, in the soil with a 1% addition of sewage sludge. The probability of these bacteria in our soil samples is high, as *Bacillu* s bacteria are typically found in the root system of maize. Based on this, we hypothesize that their abundance increased in the soil enriched with 1% SS addition and positively influenced maize growth (Fig. 2B). Fungi can compete with bacterial community for nutrients sources and therefore create a stress condition and eliminate their abundance.

The translocation of heavy metals and elements in plants is generally provided by conductive plant tissues, xylem and phloem. According to *Namdari et al. (2024)*, maize accumulates a significant fraction of heavy metals in the roots and leaves, and the translocation factor of heavy metals from leaves to maize cobs is low, thereby protecting the seeds from contamination. According to *Vymazal & Březinová (2014)*, higher concentrations of heavy metals in leaves than in stems are due to detoxification mechanisms of aboveground organs, which involve their complexation or removal from the metabolically active cytoplasm to inactive compartments, especially cell vacuoles and walls. According to them, heavy metals are accumulated in plants mainly in the order R (root) >L (leaf) >S (stem), whereas the amount of metals in leaves and stems is erratic and varies depending on the position on the stem, the length of vegetation and the season. According to *Thakur et al. (2016)*, plants get rid of heavy metals at the end of the growing season just by their accumulation in the leaves, which subsequently fall off due to senescence, thus reducing the concentrations of heavy metals in the plant body.

As a future research direction, the intention is primarily to implement a long-term field experiment as well as an economic analysis and analysis of the application of biochar and its environmental impacts. Based on our pilot study and pot experiments, distillery WWTP sludge represents a promising material that can serve as a soil additive and source of nutritionally important elements (N, P, Zn) after optimization of the pyrolysis process and pre-treatment of the input precursor to meet the criteria set for ecotoxicological safety (heavy metals, organic pollutants). For the slow pyrolysis of industrial wastewater sludge at a given temperature and direct combustion of the vapours and gases produced, a highly economic model can be proposed in further research. The process would encompass sludge drying, pyrolysis, gas and vapour combustion, gas cleaning and biochar storage. The heat from the pyrolysis gas and vapor combustion is used to provide energy for the sludge drying and pyrolysis stages. Several analyses can be carried out to assess the environmental impact, *e.g.*, Life Cycle Assessment (LCA). The intention would be to compare the relative impacts of pyrolysis and incineration of sludge from the company's industrial wastewater treatment plants on bioethanol production. In general, we could propose three potential scenarios. Namely: (I) incineration without energy recovery and subsequent landfilling of the waste produced, (II) incineration with energy recovery and subsequent landfilling of the waste produced, (III) slow pyrolysis of sludge with application of biochar as a soil additive.

## CONCLUSIONS

Physicochemical characterization of the obtained biochar showed that BC has properties typical for sewage sludge biochar, namely pH 7.71 ± 0.02; EC 1.055 ± 0.01 mS/cm; CEC 55.66 ± 2.67 cmol/kg; $C_{org}$ 36.83 ± 0.06%; $C_{inorg}$ 1.73 ± 0.06%; N 6.53%; H 1.89%; ash content 50.99%; and in terms of PAHs content of the structures, it meets the limiting concentrations (<6 mg/kg) prescribed by the EU fertilizer regulation, making it safe for land application. From the obtained results of RFA analysis of heavy metal content in BC and SS, we found that pyrolysis treatment results in multiplication of heavy metals in the obtained biochar compared to untreated sewage sludge, while the concentrations of Cu, Ni and Zn in BC exceed the permissible limits set by the EBC for biochar of classification class EBC—Agro. The determined heavy metals were represented in BC in the order Fe >Zn >Cu >Ni >Cr >Pb >Hg >Cd = As. In ecotoxicological tests with a cress seeds and earthworms, we demonstrated that there are no statistically significant changes in seed germination in a biochar-based sludge environment, but there is an increased accumulation of Zn and Cu in the tissues of soil giant nematodes. However, we confirmed by an outdoor pot experiment using maize that although there was multiplication of heavy elements in the obtained BC, a 1% addition of BC statistically significantly increased only the concentration of Ni in the plant stem compared to SS, but there was no statistically significant increase of Ni in the maize husk compared to the control. In addition, a 1% addition of BC decreases Cd concentration in flowers and Mn concentration in leaves and flowers compared to the control. In the case of heavy metal accumulation in plant weeds, despite the increased heavy metal content of BC, we rather observed a decrease in the amount of selected heavy metals and nutritionally important elements accumulated in maize weeds, which although was not statistically significant but we assume that we would observe more statistically significant differences at the increased application rate. The analysed heavy metals were accumulated in plants mostly in roots and leaves, which is a consequence of the defence mechanisms of plants against contamination of generative parts by heavy metals. The addition of BC, among others, statistically significantly reduced the total amount of microbial biomass in the soil, while increasing fungal species variability, resulting in increased P availability to plants due to the presence of a higher amount of fungal hyphae compared to SS. Based on all the results obtained, we conclude that the biochar obtained from the sewage sludge of the distillery's WWTP represents a promising material that, after optimization of the pyrolysis process and pretreatment of the input precursor in order to meet the criteria imposed for ecotoxicological safety, can serve as a soil additive and a source of nutritionally important elements.

## ACKNOWLEDGEMENTS

The authors are thankful to employees of Enviral a.s. Leopoldov for providing sludge feedstock.

### Funding

This work was funded by the Scientific Grant Agency of the Ministry of Education, Science, Research and Sport of the Slovak Republic, project number VEGA1/0399/24; The Trnava University in Trnava, project number 9/TU/2023 and the Faculty of Education of Trnava University in Trnava - project number B5/2024. This paper is a part of the dissemination activities of the project FunGlass. This project has received funding from the European Union's Horizon 2020 research and innovation programme under grant agreement number 739566. The funders had no role in study design, data collection and analysis, decision to publish, or preparation of the manuscript.

### Grant Disclosures

The following grant information was disclosed by the authors:
Scientific Grant Agency of the Ministry of Education, Science, Research and Sport of the Slovak Republic: VEGA1/0399/24.
Trnava University in Trnava: 9/TU/2023.
Faculty of Education of Trnava university in Trnava: B5/2024.
the European Union's Horizon 2020 research and innovation programme: 739566.

### Competing Interests

The authors declare there are no competing interests.

### Author Contributions

- Vladimír Frišták conceived and designed the experiments, performed the experiments, analyzed the data, prepared figures and/or tables, authored or reviewed drafts of the article, and approved the final draft.
- Lucia Polt'áková performed the experiments, analyzed the data, prepared figures and/or tables, authored or reviewed drafts of the article, and approved the final draft.
- Gerhard Soja conceived and designed the experiments, performed the experiments, analyzed the data, prepared figures and/or tables, authored or reviewed drafts of the article, and approved the final draft.
- Hana Kaňková performed the experiments, prepared figures and/or tables, authored or reviewed drafts of the article, and approved the final draft.
- Katarína Ondreičková performed the experiments, prepared figures and/or tables, authored or reviewed drafts of the article, and approved the final draft.
- Elena Kupcová performed the experiments, prepared figures and/or tables, authored or reviewed drafts of the article, and approved the final draft.
- Martin Pipíška conceived and designed the experiments, analyzed the data, prepared figures and/or tables, authored or reviewed drafts of the article, and approved the final draft.

### Data Availability

The raw data for each element and plant part are available in the Supplementary Files.

## Supplemental Information

Supplemental information for this article can be found online at http://dx.doi.org/10.7717/peerj.18184#supplemental-information.

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
