# Peer review of "Environmental risks and agronomic benefits of industrial sewage sludge-derived biochar"

_PeerJ, doi:10.7717/peerj.18184_

## Round 0.1 · original submission · Minor Revisions

Dear Authors,

Please improve your manuscript as advised by the reviewers. Clearly describe the experimental design and statistical analysis. Moreover, remove typos and grammatical mistakes.

Reviewer 1 ·

Basic reporting

Refer to additional comments

Experimental design

The research question is well-defined and relevant to the field of environmental science and agriculture.
The investigation is thorough, covering various aspects of the impact of biochar on soil and plant health. The methods are described in sufficient detail to be replicated by others.

Validity of the findings

The study has a significant impact as it addresses the ecotoxicological safety of using treated sewage sludge in agriculture, which is a pressing environmental issue. The study appears to be statistically sound, with appropriate controls and statistical tests mentioned.

Additional comments

The authors have done a good job to provide evidence about the safety and efficacy of using biochar produced by distillery wastewater treatment plan sludge as a soil amendment. There are few suggestions I would recommend that can improve the overall paper quality.

1) Define technical terms and abbreviations when first introduced to make the paper more accessible to readers who may not be familiar with the jargon.
2) Break down complex sentences into simpler ones to improve readability. Correct minor grammatical errors. Thoroughly proofread the manuscript.
3) Include a comparison with other studies that have used different types of sludge or pyrolysis temperatures to provide context for the results.
4) Discuss the broader environmental implications of the findings, including potential risks associated with the use of biochar in agriculture.
5) Address the scalability of the biochar production process and its application in different agricultural settings.
6) Include an economic analysis to assess the cost-effectiveness of using biochar as a soil additive.
7) Suggest conducting long-term field studies to assess the lasting effects of biochar application on soil health and crop yield.

·

Basic reporting

Professional language can be improved to remove repetitive points, making points clearer and more concise with shorter sentences.

No comment on the other aspects.

Experimental design

Statistical analysis approach is not mentioned in the Methods section.

No comment on the aspects.

Validity of the findings

Some improvements can be made to provide more details on study design for certain experiments.

No comment on other aspects.

Additional comments

Overall, the study provides data that for a knowledge gap in this field. The objective, introduction, results and discussion was clearly stated with adequate references to other work in the field. The study design could use more details for certain experiments. Professional language can be improved in terms of making points clearer and more concise. Detailed comments can be seen in the attachment.

·

Basic reporting

Please make your points clear by using short sentences. Don’t repeat an aspect and improve the grammar and overall language of the manuscript.

Experimental design

Please clearly explain the experimental design and the statistical analysis used during your study.

Validity of the findings

Please clearly explain the experimental design and the statistical analysis used during your study.

Additional comments

• The abstract lacks background knowledge. Please add background knowledge in the abstract.
• Line 25: What is meant by WWTP?
• The abbreviations for sewage sludge feedstock (SS) and sludge-derived biochar (BC) are inappropriate. Please improve the all abbreviations.
• Please explain a term before using the respective abbreviation.
• The results and methodology in the abstract are not clear. Please rewrite the abstract.
• The conclusion and future abstract also show ambiguity. Again write it with more clarity.
• Line 42-43: What is the specification for the used Nine containers?
• Why sand was used as growth media?
• Please explain the studied volatile compounds, gaseous contaminants, water-soluble contaminants, and direct contact of seeds first time.
• Line 155: Write the specification of the garden substrate.
• Please maintain the uniformity for mentioning of crops names like, maize (Zea mays L.) or corn.
• The results section is not properly written. It should be divided into sub-headings according to M%M.
• Correlate and compare your findings with other studies, and improve the discussion.
• Inform about the scope of this study. Inform about the possible benefit of your findings to society.
• The conclusion is very lengthy. Please delete irrelevant details from the conclusion.
• Write the limitations of this study in the conclusion section.
• Improve the language of the manuscript.

Reviewer 4 ·

Basic reporting

The manuscript provides a detailed investigation into the thermochemical treatment of sewage sludge and its potential use as a soil amendment. The study evaluates changes in chemical properties, including pH, electrical conductivity (EC), ash content, cation exchange capacity (CEC), carbon and nitrogen content, polycyclic aromatic hydrocarbons (PAHs), and heavy metals. Additionally, it examines the impact on soil microbial diversity, plant growth, and ecotoxicological effects. The manuscript provides limited specific data on microbial community changes and their functional implications for soil health. The potential negative impacts on beneficial soil microorganisms are not fully explored.

Experimental design

he discussion often lacks detailed exploration of the practical implications of the findings. For example, how changes in pH, EC, and ash content affect soil fertility and plant growth in the long term are not adequately addressed. The potential health risks associated with heavy metal accumulation in crops and their transfer through the food chain are not sufficiently discussed. A more detailed risk assessment would enhance the comprehensiveness of the study. While the discussion on PAHs is relevant, it would benefit from more specific details on the types and concentrations of PAHs detected and their potential release under different environmental conditions.

Validity of the findings

The discussion on the effects of biochar and sewage sludge on soil properties, plant growth, and microbial communities is comprehensive. The references to specific studies (e.g., Rout et al., 2015; Ason et al., 2015) provide strong support. The variability in plant responses to different biochar and sludge amendments is not fully explored. The potential negative impacts on soil health and crop yield under various environmental conditions are not sufficiently discussed. Long-term field trials and studies on a broader range of crops and soil types would help to better understand the practical implications of biochar and sewage sludge amendments. The discussion on microbial diversity and the potential ecotoxicological effects of biochar is relevant and supported by appropriate references (e.g., Wang et al., 2020; Brtnicky et al., 2019). The discussion could be enhanced by providing more specific data on microbial community changes and their functional implications for soil health. The potential for biochar to affect beneficial soil microorganisms is not fully addressed. Further research on the mechanisms by which biochar affects microbial communities and the development of strategies to mitigate any negative impacts would be beneficial.

Additional comments

Future studies should focus on the long-term stability of pH changes, EC, and ash content in biochar-amended soils and their effects on different soil types and crops. Practical guidelines for optimizing pyrolysis conditions to balance these properties should be developed. Conducting a more detailed risk assessment of heavy metal accumulation in crops and their potential transfer through the food chain would provide valuable safety data. Additionally, methods to reduce PAH content in biochar or mitigate their release should be investigated. Further research should focus on the mechanisms by which biochar affects microbial communities and the development of strategies to mitigate any negative impacts. Long-term field trials on a broader range of crops and soil types would provide more comprehensive insights

---

## Round 0.2 · accepted · Accept

The authors have addressed all of the reviewers' comments. This manuscript is ready for publication.

Reviewer 1 ·

Basic reporting

No comment

Experimental design

No comment

Validity of the findings

No comment

Additional comments

I am pleased with the revisions and the responses the authors have made to my suggestions for improving the manuscript. I accept the changes and endorse the manuscript for publication.

·

Basic reporting

Clear and unambiguous, professional English used throughout.

Experimental design

Research question well defined, relevant & meaningful. It is stated how research fills an identified knowledge gap.

Validity of the findings

All underlying data have been provided; they are robust, statistically sound, & controlled